# Automated Arrhythmia Detection Based on RR Intervals

**DOI:** 10.3390/diagnostics11081446

**Published:** 2021-08-10

**Authors:** Oliver Faust, Murtadha Kareem, Ali Ali, Edward J. Ciaccio, U. Rajendra Acharya

**Affiliations:** 1Department of Engineering and Mathematics, Sheffield Hallam University, Sheffield S1 1WB, UK; Murtadha.K.Kareem@student.shu.ac.uk; 2Sheffield Teaching Hospitals NIHR Biomedical Research Centre, Sheffield S10 2JF, UK; ali.ali@sheffield.ac.uk; 3Department of Medicine—Cardiology, Columbia University, New York, NY 10027, USA; ciaccio@columbia.edu; 4School of Engineering, Ngee Ann Polytechnic, Singapore 599489, Singapore; aru@np.edu.sg; 5Department of Bioinformatics and Medical Engineering, Asia University, Taichung 41354, Taiwan; 6School of Science and Technology, Singapore University of Social Sciences, Clementi 599494, Singapore

**Keywords:** arrhythmia detection, heart rate, RR interval, atrial fibrillation, atrial flutter, deep learning, residual neural network, detrending

## Abstract

Abnormal heart rhythms, also known as arrhythmias, can be life-threatening. AFIB and AFL are examples of arrhythmia that affect a growing number of patients. This paper describes a method that can support clinicians during arrhythmia diagnosis. We propose a deep learning algorithm to discriminate AFIB, AFL, and NSR RR interval signals. The algorithm was designed with data from 4051 subjects. With 10-fold cross-validation, the algorithm achieved the following results: ACC = 99.98%, SEN = 100.00%, and SPE = 99.94%. These results are significant because they show that it is possible to automate arrhythmia detection in RR interval signals. Such a detection method makes economic sense because RR interval signals are cost-effective to measure, communicate, and process. Having such a cost-effective solution might lead to widespread long-term monitoring, which can help detecting arrhythmia earlier. Detection can lead to treatment, which improves outcomes for patients.

## 1. Introduction

In 2015, the United Nations reported that the world population is, on average, aging [1]. It is predicted that the number of people older than 60 years will grow from 901 million to 1.4 billion by 2030 and will have doubled to 2.1 billion by 2050 [1]. As humans age, the cardiovascular system weakens, and it becomes more susceptible to disease [2]. Moreover, the arteries stiffen, and the left ventricular muscle wall thickens [3], reducing muscle compliance and affecting function adversely. The accompanying structural and electrical changes in the heart increase the risk of arrhythmia development [3]. As such, arrhythmias are abnormal rhythms of the heartbeat. These abnormal rhythms can be harmless, but some of them are critical. The most frequent type of arrhythmia is AFIB, which is manifested by uncoordinated atrial activation due to the development of a critical number of ectopic foci that initiate electrical stimuli independent of the SAN [4]. The AVN receives electrical stimuli from the atria at irregular intervals, conducting these stimuli to the ventricles, which results in an irregular QRS complex and pulse. Re-entry occurs when an impulse fails to die out after normal activation of the heart and continues to re-excite the heart. The greater the number of ectopic foci, the higher the risk of reentry, which underpins the progression from paroxysms of AFIB, to chronic AFIB [5,6]. The ECG rhythm of AFIB is chaotic and fast, at 150–220 beats per minute. Characteristically, AFIB has an abnormal RR interval, irregular rapid ventricular contraction, and the absence of a P wave in ECG. AFL occurs in a macroreentrant circuit and has a typical underlying electrophysiologic mechanism [7]. The electrical circuit in the atrium is circular and conducts rapidly leading to an atrial contraction rate of between 240 and 360 beats per minute, which gives a replicating, sawtooth waveform on ECG monitoring called a flutter wave. The AVN may transmit impulses to the ventricles regularly or irregularly, meaning that the pulse in AFL can be regular or irregular [8]. There is often significant overlap between AFIB and AFL [5,6]. Both conditions have an impact on morbidity and mortality independent of one another.

Currently, ECG measurements constitute the standard way for collecting evidence, which underpins the diagnosis of AFIB or AFL. The ECG signals could be measured as part of a screening regime that is aimed at stroke survivors, or they could be measured for symptom investigation, such as to increase our knowledge about palpitations. As such, ECG documents the electrical activity of the human heart by recording the heart polarization vector [9]. The signal is captured by placing electrodes on the human body via standardized measurement protocols. In general, more electrodes will result in a higher signal-to-noise ratio, which usually results in a better signal quality [10]. A good ECG signal quality is needed to analyze cardiac activity [11]. In order to extract disease relevant information, it is necessary to detect unambiguous signal features that describe the heartbeat. These features are the P wave, QRS complex, and T wave [12]. It is difficult to differentiate between AFIB and AFL because ECG features are often similar. Unexpected artifacts and faint manifestation of symptoms might lead to misclassifying the rhythm or overlooking important sections. That leads to intra- and inter-observer variability. CAD may be a feasible technique to reduce that variability and to limit tedious signal analysis. It might also improve preselecting signal sequences for human interpretation. Furthermore, acquisition and analysis of ECG signals requires significant data storage capacity. The ability to differentiate between AFIB and AFL using RR interval analysis will allow us to record longer signal traces for automated rhythm analysis. This will increase both the detection rate and diagnosis accuracy.

With this paper, we present a technical solution that automates AFIB and AFL detection based on RR intervals. That technical solution takes the form of a signal processing system, which uses AI for medical decision support. The system was designed and tested with benchmark data, and the results were established according to the rules of 10-fold cross-validation. During preprocessing, we ensured that training and testing data came from mutually exclusive patient groups. This ensures fully independent test sets that have not been seen during the learning phase. Hence, avoiding bias that might be introduced by training and testing the classifier with data from the same patient. The classification was done with a ResNet DL algorithm. This algorithm could differentiate AFIB, AFL, and NSR with an ACC of 97.96% and a SEN of 97.58%, as well as SPE of 98.50%. We have also established the classification performance for arrhythmia detection. For this arrhythmia/non-arrhythmia problem, the system achieved ACC = 99.98%, SEN = 100.00%, and SPE = 99.94%. Having a sound technical solution for the AFIB and AFL detection problem might lay the foundation for healthcare technology that improves outcomes for patients through longer observation duration and reduced intra- and inter-observer variability. We envision medical devices that can deliver real-time medical diagnosis by combining internet of medical things technology with advanced AI algorithms, such as the arrhythmia detection system proposed in this paper.

To support our thesis about the efficacy of the proposed arrhythmia detection method, we have structured the reminder of this manuscript as follows. The next section provides some medical background on arrhythmias. We discuss the disease symptoms and the standard measurements that are used for diagnosis. This information is relevant to appreciate the methods introduced in Section 2. In that technical part, we focus on the thought processes that gave rise to the processing structure used to train and test the deep learning network. Section 3 provides the performance measurement results for the arrhythmia detection system. These results do not stand in isolation, they were achieved by pushing the envelope of our current understanding of arrhythmia detection with physiological signals. The Discussion section highlights this point by introducing relevant research work and comparing our findings with the established knowledge. Section 5 concludes the paper with final thoughts about the work and its relevance for the medical domain.

## 2. Methods

This section outlines the methods used to support our claim that automated detection of arrhythmias in RR interval signals is possible. The methods were employed to construct a signal processing system that trains and tests a ResNet deep learning algorithm with benchmark data. Augmenting [13] and balancing [14] the dataset were two goals that guided the design strategy. Balancing a dataset means creating the same amount of training data for each class. The benchmark data for AFL had the least number of beats; hence, we employed a scrambling technique to augment the dataset. Round robin windowing was used as an augmentation technique to increase the amount of data for all signal classes. In a final step, puncturing was used to balance the dataset.

Figure 1 shows an overview block diagram of the data processing system. The processing starts with mapping the available ECG datasets from the benchmark database into three distinct classes, namely NSR, AFL, and AFIB. Subsequently, the beat-to-beat interval was extracted to form the RR interval signal. These beat-to-beat interval signals were processed such that they can be used to train and test the ResNet model. The model was evaluated with performance measures derived from a confusion matrix and ROC. The following sections introduce both data and processing steps in more detail.

### 2.1. Electrocardiogram Data

Figure 1 shows one database that sources the benchmark data to train and test the ResNet algorithm. The ‘ECG data de-noised’ (Web page: https://figshare.com/collections/ChapmanECG/4560497/2; accessed on 7 August 2021) database contained 12-lead ECG signals from 10,646 patients. A total of 4690 of the study participants were female, and the remaining 5956 were male. The following list provides the most prevalent age groups together with the relative number of participants in percent:51–60 years representing 19.82%;61–70 years representing 24.38%;71–80 years representing 16.90%.

From each patient, a 10 s ECG signal was captured with a sampling frequency of 500 Hz. The signals were measured at Chapman University and Shaoxing People’s Hospital (Shaoxing Hospital Zhejiang University School of Medicine) [15]. Each signal was labeled by a cardiologist to indicate one of 11 common rhythms. The labels came in the form of a table that links the disease label and ECG signal file name. Based on that table, we selected all ECG signal files labeled as NSR, AFIB, and AFL. The database contained 1826 NSR, 445 AFL, and 1780 AFIB signal files. Table 1 shows the number of patients for each signal class and the accumulated (over the individual patient signals within a class) ECG signal duration. As such, the table entries for ECG duration reflect the fact that all ECG signals had a length of 10 s. The ECG signal from each patient forms one data block. One such ECG data block contains an array of 12×5000 samples, where 12 indicates the number of leads and 5000 is the number of samples captured within 10 s. The term data block is used, in the description of subsequent processing steps, to denote the data from one patient. Figure 2 depicts example signals for AFIB, AFL, and NSR. There are three distinct signals for each signal class. The first of these signals depicts the 10 s ECG signal.

### 2.2. QRS Detection

The QRS detection step extracts the beat-to-beat interval from the ECG data blocks. As such, QRS is the main structural element in ECG. It is caused by ventricular depolarization that occurs when the heart muscle contracts during the heartbeat. Within the QRS complex, the R wave marks the peak, and the time location of that peak represents the time location of the heartbeat. One RR interval is the time from one R peak to the next. We have used the well-known ecg-kit, a MATLAB toolbox for ECG processing [16], for QRS detection. Within the ecg-kit framework, the wavedetect algorithm was used [11]. The resulting RR interval sequences were saved, such that the block structure was maintained. Table 1 shows the number of RR intervals for each signal class. As such, this step constitutes a significant data reduction. The following example illustrates the data reduction. There were 1826 NSR ECG data blocks, which contained 109,560,000 samples. After QRS detection, there were only 33,976 RR intervals. Hence, the compression ratio achieved by the QRS detection step was 3224.6291.

Figure 2 shows the extracted RR intervals for the example signals. The *y*-axis scale indicates the RR interval duration, and the *x*-axis scale indicates the RR interval location, i.e., the time location where the RR interval ends. Based on visual inspection, it seems that the QRS detection algorithm has inserted an additional beat for the AFIB example signal. We have highlighted the RR interval with a black circle in Figure 2.

### 2.3. Data Partitioning and Patient Scrambling

Ten-fold cross-validation involves dividing the available data into 10 parts of approximately equal size [17]. The parts were created by splitting the data along RR interval blocks. This strategy is equivalent to generating the parts along subjects. In other words, the data from one specific patient can only be found in one part. Table 2 documents this activity by reporting the number of RR intervals for NSR, AFIB, and AFL.

In all parts, the number of AFL RR intervals is more than three times lower when compared to AFIB. To adjust that imbalance, we have used patient scrambling to augment the AFL data. The patient scrambling concept is based on the fact that the part generation algorithm uses the order in which the RR interval block appears in the dataset to establish the part data. This order impacts on the data vectors, which were created through round robin windowing (see Section 2.5), because the window length is longer than the amount of RR intervals in any particular data block. Each data vector contains 100 detrended RR intervals from different patients, as outlined in the next section. Hence, a different sequence of patients in the part will result in different vectors after the windowing. In the scrambling step, we use this property to generate more AFL data. To be specific, we generated three permutations of the sequence in which the individual patient data appeared in the training and testing datasets for each part. AFL${}_{\mathrm{SC}}$ was the result of these efforts. Table 2 shows that the number of RR intervals for AFL${}_{\mathrm{SC}}$ is exactly three times greater than the number of AFL RR intervals for the same part.

### 2.4. Detrending

Detrending removes the DC offset from RR interval signals. Applying that processing method benefits the deep learning step by reducing both required network complexity and training time [18]. For our study, we have used the detrending and low-pass filter proposed by Fisher et al. [19]. The filter combination is based on an Ornstein–Uhlenbeck third-order Gaussian process, which acts on the RR interval signal directly [20,21]. After detrending, the datasets contain RR_DT samples. Table 1 shows the number of RR_DT samples. As such, the detrending step does not change the amount of data, hence the number of RR_DT samples is the same as the RR intervals. Figure 2 shows the detrended version of the RR (s) signal for each signal class. The signal graphs show that the DC bias is significantly reduced.

### 2.5. Round Robin Windowing and Puncturing

Round robin windowing augments the data by generating one data vector with 100 elements for each RR_DT sample. That method increases the data volume 100-fold. The vectors were generated by subjecting the class specific data for each part to a window length of 100. This window was slid over the RR_DT signal one sample at a time. Round robin refers to the fact that the first 100 RR_DT samples for each dataset were copied at the end, before applying the window. That extension allows us to create one data vector for each RR_DT sample.

After windowing, we have used puncturing to adjust the data size for both AFIB and AFL${}_{\mathrm{SC}}$ datasets. The puncturing algorithm removes equidistant data vectors. This technique ensures that the number of training data for each of the three classes in a part is equal. Table 3 shows that NSR has the lowest number of data vectors in any given part when compared with AFIB and AFL${}_{\mathrm{SC}}$. Therefore, we have used the number of NSR data vectors as a target for puncturing AFIB and AFL${}_{\mathrm{SC}}$. To be specific, the puncturing algorithm will reduce the number of data vectors such that it is the same as the number of NSR data vectors in the same part. For example, the number of NSR data vectors in Part 1 is 2015. After puncturing, the number of AFL${}_{P}$ and AFIB${}_{P}$ is equal to the number of NSR data vectors. The data vectors NSR, AFL${}_{P}$, and AFIB${}_{P}$ were used to train the network. The data vectors AFL${}_{\mathrm{SC}}$, NSR, and AFIB were used to test the network.

### 2.6. ResNet 10-Fold and Cross-Validation

Overfitting is the main problem for physiological signal classification with deep learning. The term refers to the fact that the deep learning network can memorize the signals itself rather than the signal properties that indicate disease symptoms. In practice, overfitting occurs when the model classifies training data correctly but fails to do so with testing data. There are a range of techniques to avoid or at least reduce overfitting. Model selection plays an important role in that process. For this study, we followed the findings by Fawaz et al. [22]. In their review on deep learning for time series classification, they found that ResNet outperforms all the other tested deep learning models. Figure 3 shows the data flow structure used to establish the ResNet model. The data flow diagram is composed from standard components that have a direct correspondence in the Python API Keras [23] for the Deep learning framework tensorflow [24]. The data flow structure shows three shortcut connections that allow information to skip the processing block. Such a structure is known as residual block. This structure can be used to address another limitation of deep learning models, namely the vanishing/exploding gradient problem [25]. From a practical perspective, this problem occurs when more network layers result in lower training accuracy, and therefore, this problem category is distinct from the general overfitting problem.

Once the network is selected, the hyperparameters need tuning. We have used a trial-and-error method to narrow down the optimal parameters. To be specific, we adopted an interactive process, which was governed by a growing understanding of the interplay between signal processing and the classification model.

Table 4 provides the number of data vectors used to train and test the ResNet. In a final step, these data vectors were used to form 10-folds that contain train and test sets. Table 3 provides the parameters for these datasets. They were arranged by selecting one part for testing and using the data vectors in the remaining parts for training. That process was repeated until every part was used for testing. The fact that the number of class-specific data vectors is equal within each part (see Table 4) leads to perfectly balanced training datasets. That means for any given Test part, the number of data vectors for NSR, AFIB, and AFL is the same. To document the data arrangement process, Column 1 in Table 4 indicates the fold and the remaining columns on the right indicate the number of training and testing data vectors. For example, for fold 1, part 1 was used for testing, and hence, parts 2 to 10 were used for training. Based on that setup, the amount of training and testing data follows from the number of class specific data vectors in each part, as provided in Table 4. For fold 1, the network was trained with all the data vectors (53,823) from AFL${}_{\mathrm{SC}}$ (17,941), AFL${}_{P}$ (17,941), and AFIB${}_{P}$ (17,941). The network was tested with all data vectors in part 1 (8275): AFL${}_{\mathrm{SC}}$ (2226), NSR (2015), and AFIB (2651).

### 2.7. Result Analysis Methods

The result analysis starts with establishing the confusion matrix based on the validation results. Table 5 defines the confusion matrix in terms of the number of beats with a true and a predicted label: $N_{\mathrm{predicted}\;\mathrm{label},\;\mathrm{true}\;\mathrm{label}}$. The predicted label was established with the ResNet classification model. Combining the 3 predicted and 3 true labels results in a confusion matrix with 3×3=9 beat labels. Table 5 shows the arrangement of these beat labels in the confusion matrix.

Both AFIB and AFL are arrhythmias. Therefore, it is reasonable to combine AFIB and AFL beats to form an arrhythmia class. The NSR beats form a non-arrhythmia class. The confusion matrix in Table 6 reflects these considerations.

Based on the confusion matrix (cm), we define TP, TN, FP, and FN for a specific class (cl) as follows:(1)$$\begin{array}{cc} {TP}_{cl} & =N_{cl,cl} \end{array}$$(2)$$\begin{array}{cc} {TN}_{cl} & =\left(\sum_{i\in Class\;set}N_{i,i}\right)-N_{cl,cl} \end{array}$$(3)$$\begin{array}{cc} {FP}_{cl} & =\left(\sum_{i\in Class\;set}N_{i,cl}\right)-N_{cl,cl} \end{array}$$(4)$$\begin{array}{cc} {FN}_{cl} & =\left(\sum_{i\in Class\;set}N_{cl,i}\right)-N_{cl,cl} \end{array}$$
where cl∈{Classset} and Classset is either {AFIB,AFL,NSR} or {Arrhythmia, Non-arrhythmia}.

These definitions were used to establish the performance measures of ACC, SPE, and SEN for the individual class: (5)$$\begin{array}{cc} {ACC}_{cl} & =\frac{{TP}_{cl}+{TN}_{cl}}{{TP}_{cl}+{TN}_{cl}+{FP}_{cl}+{FN}_{cl}} \end{array}$$(6)$$\begin{array}{cc} {SEN}_{cl} & =\frac{{TP}_{cl}}{{TP}_{cl}+{FN}_{cl}} \end{array}$$(7)$$\begin{array}{cc} {SPE}_{cl} & =\frac{{TN}_{cl}}{{TN}_{cl}+{FP}_{cl}} \end{array}$$

Ten-fold cross-validation results in 10 individual performance measures. These individual performance measures were combined to establish the overall performance. For the confusion matrix, that combination took the form of accumulating the matrix elements at the same position. The confusion matrix in Table 7 reflects these considerations. Equations (Equation 1)–(7) can be used to determine the overall performance measures.

A ROC curve illustrates how the threshold level influences the diagnostic ability of a binary classifier [26]. The area under the ROC curve indicates the general performance of the classifier, i.e., an area closer to 1 indicates a better classification performance. Equations (6) and (7) were used to calculate the class-specific True Positive Rate and False Positive Rate, respectively. The micro-average is the mean of the individual results for the three classes. The macro-average is calculated by aggregating all False Positive Rates (Web page: https://scikit-learn.org/stable/auto_examples/model_selection/plot_roc.html; accessed on 7 August 2021).

## 3. Results

The results presented in this section document the classification performance of the ResNet model. To establish that performance, 10-fold cross-validation was used to train and test the model. Table 4 details the properties of the training and testing data. As such, the table shows 10 Test Folds requiring 10 separate training and testing iterations. The training was done in 50 epochs with a batch size of 16. Categorical cross-entropy [27] was used as a loss function, and Adam [28] was used as an optimizer. For each training and testing iteration, the model with the highest testing accuracy was used to establish the confusion matrix with the methods discussed in Section 2.7. The confusion matrix results are detailed in Table 5. Having established the individual confusion matrices, we are in a position to determine the overall confusion matrix, as defined in Table 7. Equations (Equation 5)–(7) were used to calculate ACC${}_{cl}$, SEN${}_{cl}$, and SPE${}_{cl}$, where cl∈{AFIB,AFL,NSR}. Calculating the performance measures for each cl results in a 3×3 matrix for each fold.

Table 8 details the classification quality measures ACC, SPE, and SEN, as well as the confusion matrix for the 10 individual folds and overall folds. As such, all the average performance measures, detailed in the last Row of Table 8, are above 95%. This indicates how well the proposed ResNet model was able to classify AFL, AFIB, and NSR RR interval signals.

From a medical perspective, the binary problem of Arrhythmia vs. Non-Arrhythmia is also important. Therefore, we have used the definition provided in Table 6 to refine the All Test Fold confusion matrix, detailed in the last row of Table 8. With that step, we have generated the two-class results, as shown in Table 9.

The ROC curve shown in Figure 4 provides a graphical representation of the classification results. The large area under the curve is a direct result of the good classification performance indicated by the performance measures stated in Table 8 and Table 9.

## 4. Discussion

The current study investigates the problem of classifying AFIB, AFL, and NSR RR interval signals. This problem has been addressed in numerous studies, which extracted information from ECG signals. The morphology of ECG signals has dominant structural elements, such as the QRS complex, which aids the classification efforts. Human experts use distinct changes in ECG morphology for arrhythmia diagnosis. These structures are stripped away during QRS detection, which is used to extract the RR interval sequence. The RR interval reflects only the rhythm with which the heart beats. That rhythm is distinct for NSR and AFIB. Therefore, arrhythmia research based on RR interval sequences has focused on differentiating AFIB and NSR. Only Ivanovic et al. [29] address the three-class problem of AFIB, AFL, and NSR. Direct competition with this study is difficult because the authors have used a private dataset. To be specific, we could not apply the ResNet algorithm to their dataset, and therefore, we can only compare the performance results achieved with different datasets. A numerical comparison reveals that the LSTM-based detection method, proposed by Ivanovic et al., has a ≈10% lower accuracy compared to our ResNet approach. Table 10 provides an overview of arrhythmia detection studies based on ECG and RR interval signals.

In general, ECG-based arrhythmia detection achieves better accuracy values when compared to RR interval-based detection. We believe that this holds true, even though a direct comparison is not possible because different datasets were used to establish the performance results. ECG holds all the information about the electrical activity of the human heart. As such, the RR interval is part of this information. Hence, during the process of extracting the RR intervals, we lose all information contained in the morphology of the ECG. However, when we compare the accuracy performance reported by Fujita et al. [31] with the ResNet accuracy, we find that our performance is just 0.49% lower. The small performance benefit might not justify the increased measurement effort and significantly higher data rate of ECG signals when compared to RR interval signals. The increased measurement effort results in the fact that ECG monitors require expert instrumentation, i.e., the sensors must be attached by a qualified nurse. In contrast, RR intervals can be measured with sensors that were placed by patients [45]. State-of-the-art ECG sensors deliver 250 samples per second. In contrast, the heart beats around once a second, producing about one RR interval value per second. The fact that RR interval signals have a 250 times lower data rate when compared to ECG signals leads to significant cost savings when it comes to communication, storage, and processing [46].

RR interval-based arrhythmia detection becomes even more important when we move away from the electrical activity of the human heart and consider RR intervals extracted from pulse signals [47]. Pulse sensors are less expensive and more readily available when compared to RR interval sensors that measure the electrical activity of the human heart [48]. Therefore, pulse sensors can be used in wearable devices, such as smart watches. Coupled with the low data rate of RR interval signals, wearable technology may facilitate low barrier and low-cost arrhythmia detection systems. Such systems are governed by the laws of big data, where individual beat classifications become less significant when compared to accumulated evidence. Furthermore, big data helps to diversify and to improve classification results. This may lead to a better understanding and detection of early-stage arrhythmia. Fuzzy logic might play a role to support the analysis task by mitigating uncertainties and reducing inaccuracies [49,50,51].

### 4.1. Limitations

The ECG signals were measured from a large number of patients. However, the signal duration is only 10 s. Longer data sequences are needed to validate and potentially redesign arrhythmia detection functionality. Associated with the available signal source is another limitation of the study, namely the RR intervals were extracted from ECG. The process of establishing the RR intervals is likely to be different in a practical setting because of the economic cost and the infrastructure requirements of ECG recording. To be specific, cost-effective heart rate monitors, such as sensors worn on chest and wrist, use different methods to establish the RR intervals. Hence, testing out the data acquisition is required before our results can be used to create practical systems that improve clinical practice.

The suspected surplus RR interval shown in Figure 2 provides a poignant reminder that errors can occur during physiological signal processing. QRS detection is no exception to that rule, and therefore, RR interval signals may contain errors. Correcting these errors through visual inspection by a human expert is impractical, and it would distort the signal interpretation results. Impracticality follows directly from the large amounts of data that would need to be verified. That verification process would significantly diminish the feasibility of any problem solution based on RR interval signals. Hence, a practical arrhythmia detection system must be able to cope with errors in the RR interval signals, and the ability to do so should be documented during the design time. Therefore, it would be counterproductive to remove these errors from the training and testing data. Hence, we require that a signal interpretation method, such as the proposed arrhythmia detector, must deliver robust results even in the presence of error. This robustness can only be ensured by keeping the error in the RR interval signals.

### 4.2. Future Work

In this paper, we showed that a ResNet model can discriminate between AFL, AFIB, and NSR RR interval signals. In the future, we need to determine how the ResNet model performs in a practical medical decision support scenario. Such a study could provide deeper insight into the role of benchmark data for arrhythmia detection. However, more and longer measurement data are needed to address the limitations outlined in the previous section. Fuzzy logic [52,53] for QRS detection might help to reduce errors and thereby improve the practical relevance of the proposed arrhythmia detection method.

## 5. Conclusions

With this study we showed that the presence of AFIB and AFL manifests itself in RR interval signals. The medical need for this study comes from the fact that arrhythmia increases the risk of morbidity and mortality due to AFIB-related complications, such as stroke. Currently, most arrhythmias are detected based on manual interpretation of ECG signals. This process is time-consuming and expensive, which limits both the number of observations and the observation duration. To detect more arrhythmias, it is necessary to address both shortcomings with a cost-effective solution. We put forward that RR interval measurements could underpin such a cost-effective detection solution because these signals require only a simple measurement setup and have a low data rate. Hence, detection systems based on RR intervals have the potential to be significantly less expensive when compared to ECG-based arrhythmia detection. However, computer support for RR interval signal-based arrhythmia detection is mandatory, whereas computer support for ECG-based arrhythmia detection is optional, but it is desired to reduce the cost.

Our ResNet deep learning algorithm can discriminate AFIB, AFL, and NSR with ACC = 99.98%, SEN = 100.00%, SPE = 99.94%. These results were obtained with 10-fold cross-validation. The fact that the performance was similar over all folds supports our claim that the developed algorithm is robust. This robustness is important when we transit from the theoretical setting in the data science lab to practical applications in a clinical setting. In such a clinical setting, the proposed algorithm becomes an adjunct tool that can support a cardiologist during the diagnostic procedure. We envision a two-stage diagnostic process where machine algorithms and human experts work cooperatively to achieve good outcomes for patients. To be specific, the proposed deep learning algorithm can be used to analyze RR interval signals in real time. The human practitioner verifies the detection result and thereby establishes a diagnosis. That approach utilizes the diligence of the deep learning algorithm and the ability of human experts to combine knowledge about the patient together with measurement evidence to reach a sound diagnosis.

## Figures and Tables

**Figure 1 diagnostics-11-01446-f001:**
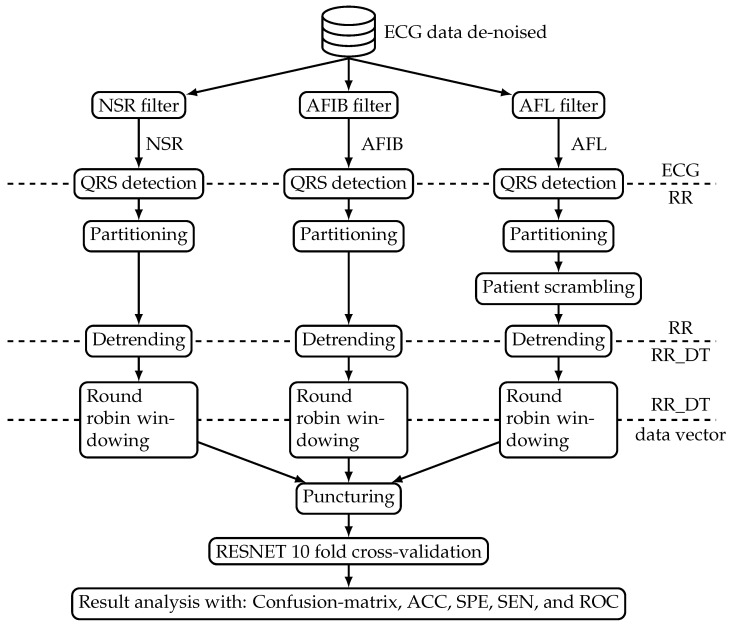
Block diagram of the study setup.

**Figure 2 diagnostics-11-01446-f002:**
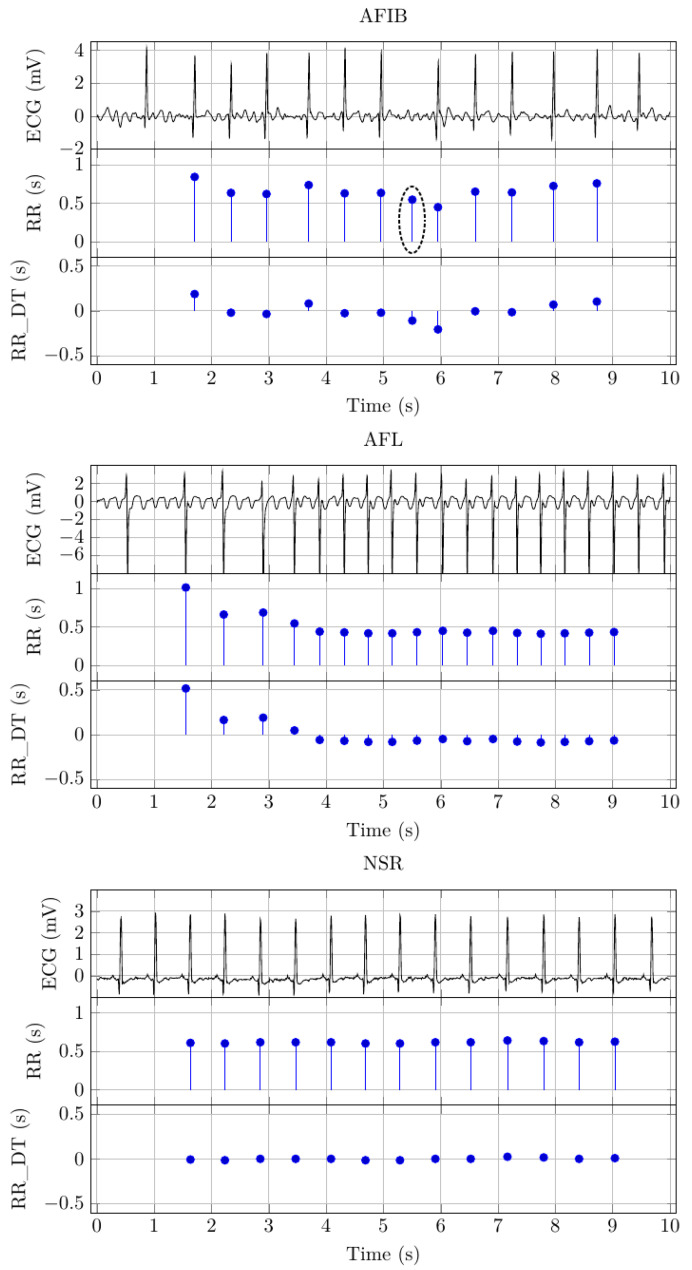
Example plots from AFIB, AFL, and NSR signal classes. The ECG signal was measured with the aVL lead. The RR intervals, plotted as RR intervals over time, were derived from the ECG via QRS detection. The detrended RR intervals were plotted as RR_DT over time. Visual inspection indicates that the AFIB RR (s) signal shows an additional beat, which has been encircled with a dashed ellipse.

**Figure 3 diagnostics-11-01446-f003:**
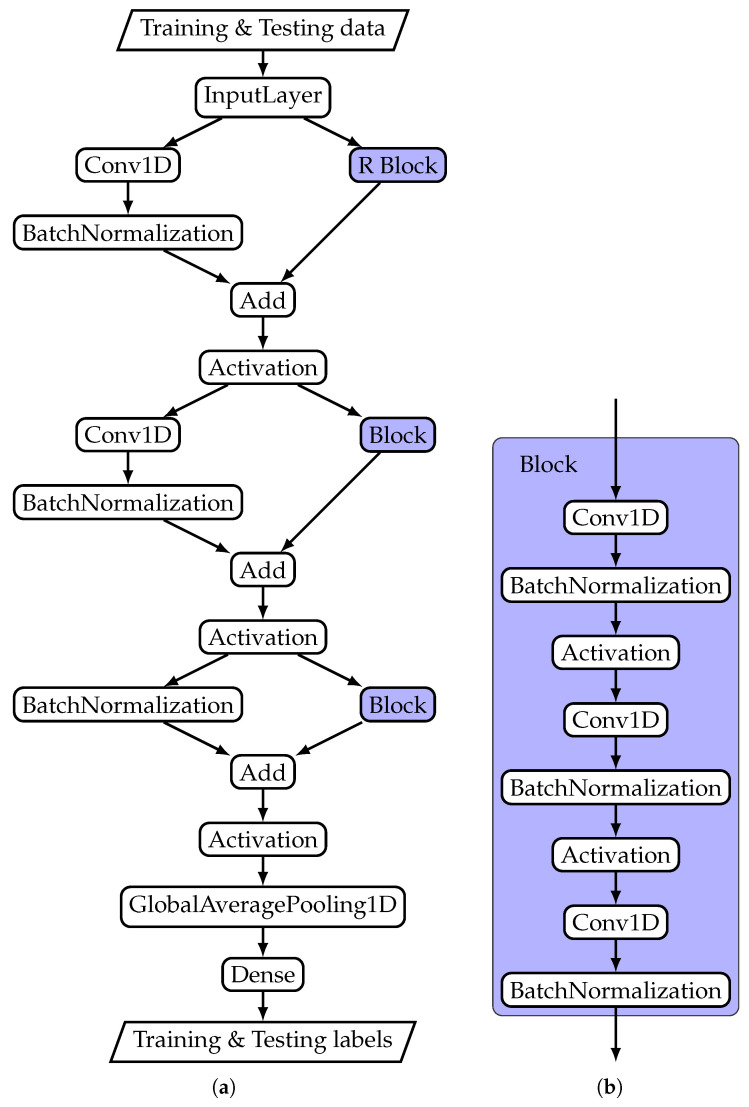
ResNet structure used for training and testing: (**a**) Network super structure; (**b**) Block structure.

**Figure 4 diagnostics-11-01446-f004:**
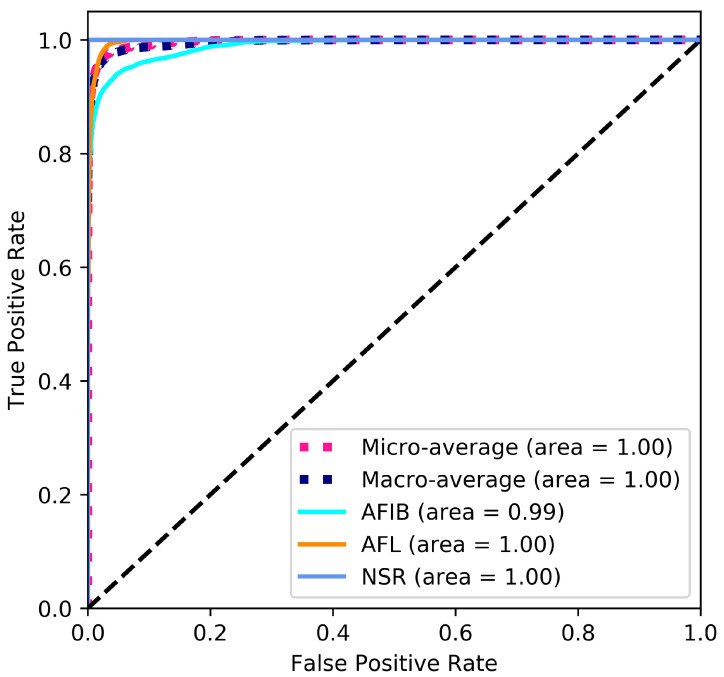
ROC curve based on the results from all 10 folds.

**Table 1 diagnostics-11-01446-t001:** Data properties for the three signal classes. The ‘ECG Duration (s)’ column provides the time duration of all ECG signal blocks for each individual class. After that, the two columns to the right provide the number of RR intervals and the number of RR_DT samples, respectively. The last two columns on the right provide the number of blocks and number of patients for each signal class.

	Property	ECG Duration (s)	RR Intervals	RR_DT Samples	Number of Blocks	Number of Patients
Class						
NSR		18,260	33,976	33,976	1826	1826
AFIB		17,800	25,995	25,995	1780	1780
AFL		4450	7536	7536	445	445
Total		40,510	67,507	67,507	4051	4051

**Table 2 diagnostics-11-01446-t002:** The number of RR intervals per signal class in each part. AFL${}_{\mathrm{SC}}$ denotes the scrambled AFL dataset.

	Part	1	2	3	4	5	6	7	8	9	10
Class											
NSR		2015	1980	1980	2029	2020	1973	1992	2017	1975	1975
AFIB		2651	2667	2584	2566	2633	2649	2594	2512	2604	2535
AFL		742	759	786	784	762	766	727	702	721	787
AFL${}_{\mathrm{SC}}$		2226	2277	2358	2352	2286	2298	2181	2106	2163	2361

**Table 3 diagnostics-11-01446-t003:** The number of data vectors per signal class in each part. AFL${}_{P}$ and AFIB${}_{P}$ denote the punctured datasets for NSR and AFIB, respectively.

	Part	1	2	3	4	5	6	7	8	9	10
Class											
AFIB		2651	2667	2584	2566	2633	2649	2594	2512	2604	2535
AFL${}_{\mathrm{SC}}$		2226	2277	2358	2352	2286	2298	2181	2106	2163	2361
NSR		2015	1980	1980	2029	2020	1973	1992	2017	1975	1975
AFL${}_{P}$		2015	1980	1980	2029	2020	1973	1992	2017	1975	1975
AFIB${}_{P}$		2015	1980	1980	2029	2020	1973	1992	2017	1975	1975

**Table 4 diagnostics-11-01446-t004:** The number of data vectors used for training and testing during 10-fold cross-validation.

Fold	Training Data				Testing Data			
	NSR	AFIB	AFL	Total	NSR	AFIB	AFL	Total
1	17,941	17,941	17,941	53,823	2015	2651	2226	6892
2	17,976	17,976	17,976	53,928	1980	2667	2277	6924
3	17,976	17,976	17,976	53,928	1980	2584	2358	6922
4	17,927	17,927	17,927	53,781	2029	2566	2352	6947
5	17,936	17,936	17,936	53,808	2020	2633	2286	6939
6	17,983	17,983	17,983	53,949	1973	2649	2298	6920
7	17,964	17,964	17,964	53,892	1992	2594	2181	6767
8	17,939	17,939	17,939	53,817	2017	2512	2106	6635
9	17,981	17,981	17,981	53,943	1975	2604	2163	6742
10	17,981	17,981	17,981	53,943	1975	2535	2361	6871

**Table 5 diagnostics-11-01446-t005:** The confusion matrix for AFIB, AFL, and NSR.

		Predicted Label		
		**AFIB**	**AFL**	**NSR**
	**AFIB**	$N_{\mathrm{AFIB},\;\mathrm{AFIB}}$	$N_{\mathrm{AFL},\;\mathrm{AFIB}}$	$N_{\mathrm{NSR},\;\mathrm{AFIB}}$
**True Label**	**AFL**	$N_{\mathrm{AFIB},\;\mathrm{AFL}}$	$N_{\mathrm{AFL},\;\mathrm{AFL}}$	$N_{\mathrm{NSR},\;\mathrm{AFL}}$
	**NSR**	$N_{\mathrm{AFIB},\;\mathrm{NSR}}$	$N_{\mathrm{AFL},\;\mathrm{NSR}}$	$N_{\mathrm{NSR},\;\mathrm{NSR}}$

**Table 6 diagnostics-11-01446-t006:** The confusion matrix for arrhythmia and non-arrhythmia.

		Predicted Label	
		**Arrhythmia**	**Non-Arrhythmia**
	**Arrhythmia**	$N_{\mathrm{AFIB},\;\mathrm{AFIB}}+N_{\mathrm{AFL},\;\mathrm{AFIB}}$	$N_{\mathrm{NSR},\;\mathrm{AFIB}}$
**True Label**		$+N_{\mathrm{AFIB},\;\mathrm{AFL}}+N_{\mathrm{AFL},\;\mathrm{AFL}}$	$+N_{\mathrm{NSR},\;\mathrm{AFL}}$
	**Non-Arrhythmia**	$N_{\mathrm{AFIB},\;\mathrm{NSR}}+N_{\mathrm{AFL},\;\mathrm{NSR}}$	$N_{\mathrm{NSR},\;\mathrm{NSR}}$

**Table 7 diagnostics-11-01446-t007:** The average cross-validation confusion matrix. $\sum_{\langle \mathrm{Test}\;\mathrm{Fold}\rangle }$ indicates the sum over all Test Folds.

		Predicted Label		
		**AFIB**	**AFL**	**NSR**
	**AFIB**	$\sum_{\langle \mathrm{Test}\;\mathrm{Fold}\rangle }\left\{N_{\mathrm{AFIB},\;\mathrm{AFIB}}\right\}$	$\sum_{\langle \mathrm{Test}\;\mathrm{Fold}\rangle }\left\{N_{\mathrm{AFL},\;\mathrm{AFIB}}\right\}$	$\sum_{\langle \mathrm{Test}\;\mathrm{Fold}\rangle }\left\{N_{\mathrm{NSR},\;\mathrm{AFIB}}\right\}$
**True Label**	**AFL**	$\sum_{\langle \mathrm{Test}\;\mathrm{Fold}\rangle }\left\{N_{\mathrm{AFIB},\;\mathrm{AFL}}\right\}$	$\sum_{\langle \mathrm{Test}\;\mathrm{Fold}\rangle }\left\{N_{\mathrm{AFL},\;\mathrm{AFL}}\right\}$	$\sum_{\langle \mathrm{Test}\;\mathrm{Fold}\rangle }\left\{N_{\mathrm{NSR},\;\mathrm{AFL}}\right\}$
	**NSR**	$\sum_{\langle \mathrm{Test}\;\mathrm{Fold}\rangle }\left\{N_{\mathrm{AFIB},\;\mathrm{NSR}}\right\}$	$\sum_{\langle \mathrm{Test}\;\mathrm{Fold}\rangle }\left\{N_{\mathrm{AFL},\;\mathrm{NSR}}\right\}$	$\sum_{\langle \mathrm{Test}\;\mathrm{Fold}\rangle }\left\{N_{\mathrm{NSR},\;\mathrm{NSR}}\right\}$

**Table 8 diagnostics-11-01446-t008:** Analysis results for the individual and all folds.

Fold	cl	ACC${}_{\mathrm{cl}}$ (%)	SEN${}_{\mathrm{cl}}$ (%)	SPE${}_{\mathrm{cl}}$ (%)	Confusion Matrix		
	AFIB	97.16	92.72	99.27	2064	162	0
1	AFL	97.16	98.72	96.18	34	2617	0
	NSR	100.00	100.00	100.00	0	0	2015
	AFIB	99.87	99.60	100.00	2268	9	0
2	AFL	99.87	100.00	99.79	0	2667	0
	NSR	100.00	100.00	100.00	0	0	1980
	AFIB	95.81	87.70	100.00	2068	290	0
3	AFL	95.81	100.00	93.31	0	2584	0
	NSR	100.00	100.00	100.00	0	0	1980
	AFIB	96.95	91.11	99.93	2143	209	0
4	AFL	96.95	99.88	95.23	3	2563	0
	NSR	100.00	100.00	100.00	0	0	2029
	AFIB	98.83	96.98	99.74	2217	69	0
5	AFL	98.83	99.54	98.40	12	2621	0
	NSR	100.00	100.00	100.00	0	0	2020
	AFIB	100.00	100.00	100.00	2298	0	0
6	AFL	99.96	100.00	99.93	0	2649	0
	NSR	99.96	99.85	100.00	0	3	1970
	AFIB	96.81	90.10	100.00	1965	216	0
7	AFL	96.81	100.00	94.82	0	2594	0
	NSR	100.00	100.00	100.00	0	0	1992
	AFIB	94.32	83.05	99.56	1749	357	0
8	AFL	94.32	99.20	91.34	20	2492	0
	NSR	100.00	100.00	100.00	0	0	2017
	AFIB	98.28	95.42	99.63	2064	99	0
9	AFL	98.15	99.35	97.39	17	2587	0
	NSR	99.86	99.54	100.00	0	9	1966
	AFIB	100.00	100.00	100.00	2361	0	0
10	AFL	100.00	100.00	100.00	0	2535	0
	NSR	100.00	100.00	100.00	0	0	1975
	AFIB	97.82	93.76	99.81	21,197	1411	0
All	AFL	97.80	99.67	96.66	86	25,909	0
	NSR	99.98	99.94	100.00	0	12	19,944

**Table 9 diagnostics-11-01446-t009:** Overall classification, where cl= Arrhythmia.

ACC${}_{\mathrm{cl}}$ (%)	SEN${}_{\mathrm{cl}}$ (%)	SPE${}_{\mathrm{cl}}$ (%)	Confusion Matrix	
99.98	99.94	100.00	48,603	0
			12	19,944

**Table 10 diagnostics-11-01446-t010:** Selected arrhythmia detection studies using RR intervals and ECG. pDB used were: MIT-BIH Atrial Fibrillation Database (afdb), MIT-BIH Arrhythmia Database (mitdb), MIT-BIH Malignant Ventricular Arrhythmia Database (vfdb), Creighton University Ventricular Tachyarrhythmia Database (cudb), MIT-BIH Normal Sinus Rhythm Database (nsrdb), MIT-BIH Long Term Database (ltdb), European ST-T Database (edb), and ecgdb. Hospital data come from non-publicly accessible databases.

Author Year	Method	Data	Performance
Type	DB	Rhythm	ACC	SPE	SEN
**Current**	Detrending, ResNet	RR	ecgdb	AFIB AFL NSR	**99.98**	**100.00**	**99.94**
Faust and Acharya 2021 [30]	Detrending, ResNet	RR	ecgdb	SVT, ST, SB, AFIB, AFL, NSR	98.55	94.30	99.40
Ivanovic et al. 2019 [29]	CNN, LSTM	RR	Hospital	NSR, AFIB AFL	88		87.09
Fujita et al. 2019 [31]	CNN with normalization	ECG	afdb, mitdb, vfdb	AFIB, AFL, VFIB, NSR	98.45	99.87	99.27
Faust et al. 2018 [32]	LSTM	RR	afdb	AFIB NSR	98.39	98.32	98.51
Acharya et al. 2017 [33]	CNN with Z-score	ECG	afdb, mitdb, vfdb	AFIB, AFL, VFIB, NSR	92.50	98.09	93.13
Henzel et al. 2017 [34]	Statistical features with generalized Linear Model	RR	afdb	AFIB NSR	93	95	90
Desai et al. 2016 [35]	RQA with DecisionTree, RandomForest, RotationForest	ECG	afdb, mitdb, vfdb	AFIB, AFL, VFIB, NSR	98.37		
Acharya et al. 2016 [36]	Thirteen nonlinear features with ANOVA with KNN and DT	ECG	afdb, mitdb, vfdb	AFIB, AFL, VFIB, NSR	97.78	99.76	98.82
Hamed et al. 2016 [37]	DWT, PCA and SVM	ECG	afdb	AFIB, AFL, NSR	98.43	96.89	98.96
Xia et al. 2018 [38]	STFT/SWT with CNN	ECG	afdb	AFIB	98.63	98.79	97.87
Petrenas et al. 2015 [39]	Median filter with threshold	RR	nsrdb, afdb	AFIB NSR		98.3	97.1
Zhou et al. 2014 [40]	Median filter & Shannon entropy with threshold	RR	ltafdb, afdb, nsrdb	AFIB NSR	96.05	95.07	96.72
Muthuchudar et al. 2013 [41]	UWT NN	ECG	afdb	AFIB, VFIB, NSR	96		
Yuan et al. 2016 [42]	Unsupervised autoencoder NN Softmax regression	ECG	afdb, nsrdb, ltdb, hospital	AFIB	98.18	98.22	98.11
Dinakarrao et al. 2018 [43]	Daubechies-6 with counters Anomaly detector	ECG	mitdb	AFIB, VFIB	99.19	98.25	78.70
Salem et al. 2018 [44]	Spectogram with CNN	ECG	afdb nsrdb vfdb edb	AFIB, AFL VFIB NSR	97.23

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
