# Peer review of "Automated Arrhythmia Detection Based on RR Intervals"

_diagnostics, 2021, doi:10.3390/diagnostics11081446_

Round 1
Reviewer 1 Report
The manuscript presents a methodology for detection of atrial fibrillation and atrial flutter based on analysis of RR-interval signals via ResNet. The authors declare high accuracy results, which are better than the reported in literature. A possible bias of the results towards higher values could be their calculation over the validation set, but not over a fully independent test set, which is not seen during the learning of the network. This point should be clarified in the text of the manuscript.
Specific comments, questions and recommendations:
- On page 11 the authors have started the numbering of tables again from 1. Thus, Table 1 and Table 3 appear 2 times, while Table 2 appears even 3 times. Check the numbering of the tables and their references in the text.
- On page 11 – the 2nd equation should be TNcl instead of TPcl.
- What results are presented in Table 2 (page 12)? Are these the test results? And are these really test results on an independent data, which is not used in the training and validation of the network? Or these are the validation results? Explain on what data is performed the validation!
- Fair comparison is not possible due to the different databases. However, the authors should at least extend Table 2 (with comparative results) and provide information about the presented accuracies – i.e. were they achieved on really independent test dataset or on the validation sets that are used during the training of the model?
Reviewer 2 Report
*) The paper is interesting, well written and well structured. The methodological rigor is not lacking and the reading is more than pleasant. However, I propose some suggestions to improve its readability.
*) A greater and better description of the available database would be useful.
*) The ECG signals, as the authors surely know, can be affected by uncertainties and/or inaccuracies. Therefore, strictly speaking, to perform QRS detection it would be necessary to use techniques based on fuzzy logic. Notwithstanding that such an approach is beyond the scope of this work, I still recommend that the authors insert a sentence that highlights this possibility by putting the following relevant works in the bibliography:
doi: 10.1109/20.952655
this work, even if dated, is now a milestone in the field of fuzzy detection. The technique has been applied to a NdT problem but, given the transversality of the approach, nothing prevents it from being used for QRS detection.
doi: 10.1016/j.trf.2018.05.033
*) The classification would also deserve special attention. In other words, also in this case the fact that the signals could be affected by uncertainties and / or inaccuracies would require fuzzy classification techniques. Obviously, as for the previous remark, I recommend inserting a sentence in the text that highlights this possibility by putting the following relevant works in the bibliography:
doi: 10.1155/2014/201243 (work that presents a good classification technique based on latest generation fuzzy clustering)
doi: 10.1109/TFUZZ.2017.2728521
doi: 10.4018/IJFSA.2017040102
Round 2
Reviewer 1 Report
The authors have addressed the recommendations in my previous review report and the new version of the manuscript is suitable for publication. Next time, a version of the manuscript with highlighted changes would be highly appreciated.